# HistoBench: World History Event Extraction and Cognitive-Level Benchmarking of Generative AI

## Abstract

We present HistoBench, a benchmark and dataset designed to evaluate and improve large language models' (LLMs) ability to reason about complex, temporally grounded historical narratives. While LLMs perform well on general language tasks, their historical understanding remains limited. HistoBench provides a richly annotated collection of global events, timelines, and causal chains, alongside an interactive timeline and global map to enhance accessibility for research and education. To assess reasoning across multiple depths, we introduce a set of 1,007 historical questions structured around Bloom's Taxonomy, covering levels from factual recall (*Remember*) to higher-order reasoning (*Evaluate* and *Create*). Our results show that models perform well on spatial and entity recognition but struggle more with temporal reasoning. Among the evaluated systems, DeepSeek-V3 consistently outperforms GPT4o-mini and Gemma-3 across nearly all levels, achieving over 90% accuracy at the most advanced stages of evaluation and creation, highlighting its stronger capacity for complex historical reasoning.

## 1 Introduction

The emergence of digital humanities over the last two decades has fundamentally transformed scholarship in the humanities, particularly in the field of history (Fafalios et al., 2023). Historical documents are now massively digitized into photos and texts, allowing researchers to query across collections and languages. This digitization has created an enormous volume of archives and archival data available digitally, producing a valuable but under-utilized source of large-scale digital data for digital humanities scholars (Hawkins, 2021). However, several challenges remain in this domain.

The challenges in the historical data analysis are as follows: **(i) Under-exploration of certain historical tasks:** One of the primary challenges in digital humanities is the under-exploration of certain historical tasks, particularly event extraction, which has either been applied to small-scale datasets or constrained by limited event typologies with predefined event categories (Rovera et al., 2019) (Hervieux et al., 2024). This limitation has restricted the broader application and generalizability of event extraction methods in historical research. **(ii) The lack of structured data:** Most historical texts are not in clean, structured formats suitable for direct computational analysis, therefore requiring extensive preprocessing before being usable in NLP pipelines (Wakabayashi, 2019). Available historical texts can be divided into three types from the point of automated text analysis: initially digital, printed/written but digitized, and non-digitized printed/written texts (Huistra & Mellink, 2016). In the case of solely printed or written texts, digitization is just the first step, as digitized text must be preprocessed to make it proper for automated analysis through steps like correction of Optical Character Recognition (OCR), concept or meta tagging, and lemmatization (Szabó et al., 2020). **(iii) Presenting large historical datasets:** While large-scale analysis of historical sources can provide a broader and more nuanced understanding of historical events, the sheer volume of extracted data can be overwhelming. For it to be useful, especially to non-experts, the data must be organized, filtered, and displayed in an accessible and user-friendly format. The scale and diversity of such collections presents particular challenges in identifying and extracting relevant content (Leavy et al., 2019). **(iv) Benchmarking Gaps in Historical Knowledge Evaluation:** Evaluating large language models on historical knowledge has become a key area of research as these systems are increasingly used for educational and informational purposes (Garcia & Weilbach, 2023). History

presents unique challenges for LLMs because it requires not just memorizing isolated facts, but understanding complex relationships between events, people, and time periods (Kandpal et al., 2023). Moreover, our historical knowledge and the available digital data are heavily skewed toward Western narratives, and this Western bias is also evident in the knowledge encoded by large language models (Keleg & Magdy, 2023).

To address the first challenge, we employed large language models (LLMs) and used prompt engineering techniques to perform tasks such as historical event extraction. To tackle the second challenge, we developed a series of preprocessing steps, particularly tailored to the constraints and nuances of feeding book-length texts into LLMs. To overcome the third challenge, we designed a web-based user interface that enables users to visually explore and filter the extracted events through interactive timelines and maps. Therefore, both academic researchers and non-specialist users can benefit from the outputs. Scholars can use the platform for historical investigations across a wide range of time periods and geographic regions, regardless of their specific area of expertise. In addition, the platform serves as an educational tool, accessible to general users with an interest in learning about historical events and patterns. To address the forth gap, we curated a dataset of 1,007 multiple-choice questions derived from the structured historical data extracted from our source texts. This dataset covers a wide variety of time periods and regions, enabling a fair and representative evaluation. We then used it to benchmark the historical understanding of several state-of-the-art LLMs, providing new insights into their performance and limitations in processing historical content. Figure 1 provides a visual overview of the event extraction process and large language model (LLM) evaluation pipelines in our work.

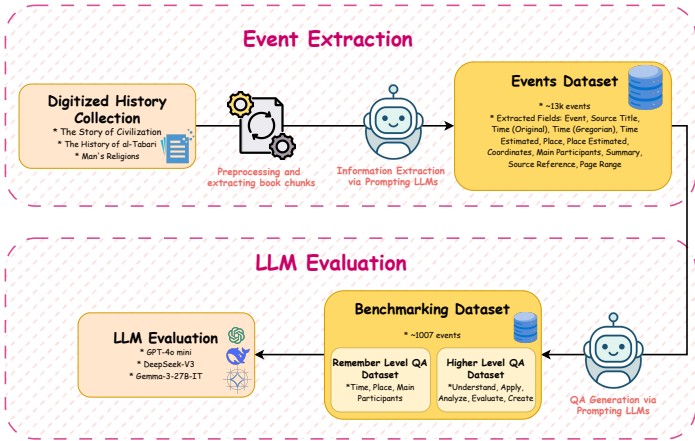

Figure 1: An overview of our pipeline for historical event extraction and evaluation. The top section illustrates how structured event data is extracted from digitized historical texts using LLMs. The bottom section shows how the resulting dataset is used for evaluating LLMs across multiple reasoning levels based on Bloom's Taxonomy.

## 2 RELATED WORK

**Event Extraction:** A common approach in the task of event extraction has been to decompose it into smaller subtasks. For example, (Nguyen & Grishman, 2018) employs Graph Convolutional Networks (GCNs) to perform event detection, which involves identifying whether a specific event occurs within a given text. Another example is GRIT (Du et al., 2021), which uses a transformer-based model to extract entities related to events.

Subsequent work in event extraction has largely framed the task as a classification problem, often focusing on identifying and categorizing event triggers—words that explicitly indicate the occurrence of an event, typically the main verb in a sentence. This approach is based on annotation guidelines such as those provided by the ACE dataset (ACE), which defines and categorizes event types. For example, Sprugnoli and Tonelli (Sprugnoli & Tonelli, 2019) introduced an annotation scheme that classifies events into 22 categories and created a dataset with these annotations, along with a model

to automate the annotation process. The BRAD dataset (Lai et al., 2021) is another relevant example. It contains annotated historical texts related to Black uprisings found in 19th-century African American newspapers. The study reported that existing models, based primarily on BERT, struggled to perform well on this dataset.

A significant shift in methodology came with research showing that framing event extraction as a question answering (QA) task yields promising results [liu-etal-2020-event]. Follow-up studies have validated the effectiveness of this approach. For instance, (Borenstein et al., 2023) introduced a multilingual dataset based on early modern colonial-era newspaper advertisements that document formerly enslaved individuals who liberated themselves. Using a QA-based approach with RoBERTa models, they achieved strong results on these historical texts.

However, these prior works have notable limitations: the questions are typically handcrafted, the tasks are limited to specific event types, and the datasets are small in scale and narrowly focused. Given the demonstrated success of QA formulations for event extraction, the emergence of large language models (LLMs) presents a powerful opportunity. These models inherently operate well in QA-like formats and enable large-scale, high-accuracy event extraction across diverse historical texts, without being constrained by fixed event taxonomies.

**Visualizing Historical Events:** In terms of visualizing historical events on a timeline, relatively few studies have addressed this challenge. Bedi et al. (Bedi et al., 2017) utilized the TimeMapper tool (https://timemapper.okfnlabs.org/) for this purpose, using the NER component of Stanford CoreNLP (Manning et al., 2014) to extract events. However, their extracted events were limited in scope, based on only around 200 sentences. Another study by Hienert et al. (Hienert & Luciano, 2012) worked with a larger dataset spanning from 300 BC to 2013. Their dataset was derived from structured data on Wikipedia, where events are already listed in chronological format on dedicated pages. Their work focused primarily on building a pipeline for event extraction and visualization from this semi-structured source.

**Historical Benchmarking for LLMs:** General-purpose evaluation benchmarks like MMLU (Hendrycks et al., 2021) are widely adopted across numerous academic domains, including history, as proxies for assessing large language models' reasoning and encyclopedic knowledge. However, these benchmarks are not tailored to the unique demands of historical reasoning: they do not offer contextual narrative structure, causal chaining, or temporally grounded evaluation specific to history, motivating the need for a domain-specific dataset.

Dedicated historical and temporal reasoning benchmarks have made important progress, but each exhibits key limitations. HiST-LLM, built from the Seshat Global History Databank, provides structured coverage of historical societies from the Neolithic to the Industrial Revolution, but emphasizes basic factual recall and lacks systematic alignment with cognitive levels like analysis or evaluation (Hauser et al., 2024). HistBench, developed alongside the HistAgent platform, offers multilingual and multimodal historical QA, yet remains limited in scale (hundreds of questions) and does not integrate Bloom's Taxonomy to balance cognitive complexity across tasks (Qiu et al., 2025). Temporal reasoning benchmarks such as TRAM (Wang & Zhao, 2024) and TimeBench (Chu et al., 2024) provide broad coverage of tasks involving ordering, duration, frequency, arithmetic, and some aspects of causality. Nonetheless, they lack support for causal-chain visualizations and structured narrative event extraction, and similarly omit a systematic approach to cognitive-level design.

In contrast, our work addresses these gaps by delivering (1) broad temporal and geographic representation of extracted events; (2) an interactive, map-based visualization interface; and (3) a deliberately designed set of 1,007 multiple-choice questions, crafted according to Bloom's Taxonomy to span remembering through creating cognitive levels. This enables more interpretable and cognitively informed evaluation of LLM historical reasoning.

Table 1: Basic quantitative statistics of the selected historical texts, including total pages, word counts, and character counts

| Book | Pages # | Words # | Characters # |
|------|--------:|--------:|-------------:|
| The Story of Civilization | 9,570 | 4.24M | 24.7M |
| The History of al-Tabari | 6,166 | 1.63M | 8.11M |
| Man's Religions | 768 | 0.30M | 1.76M |
| **Total** | **16,504** | **6.17M** | **34.6M** |

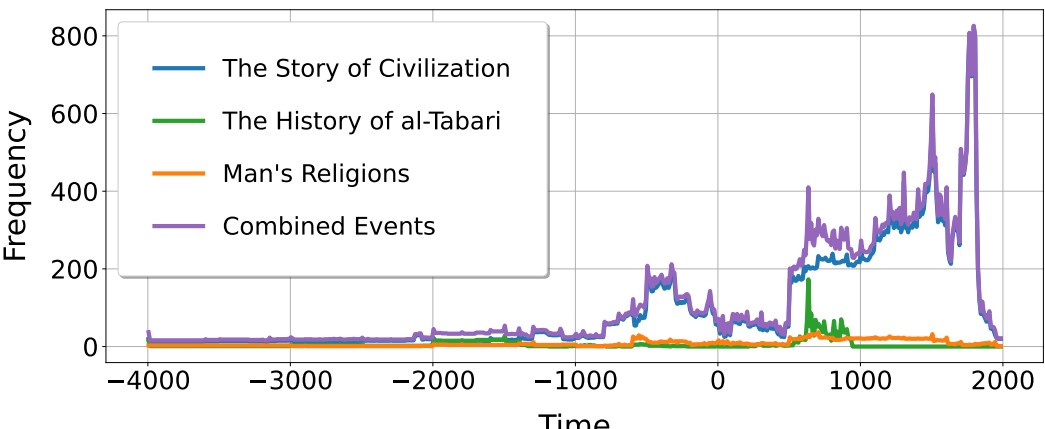

Figure 2: Temporal distribution of events in the full dataset, categorized by source texts: The Story of Civilization, The History of al-Tabari, and Man's Religions.

## 3 DATASET

### 3.1 EVENTS DATASET

We analyzed three major historical texts [1] to extract a wide range of world events, aiming to broaden the geographic and cultural scope beyond a predominantly Western focus. Some information about the size of these resources is provided in Table 1, which summarizes the number of pages, words, and characters for each book as well as their combined totals. Our resources include:

**The Story of Civilization**, an 11-volume series by Will and Ariel Durant (1935–1975), traces the broad sweep of world history from prehistoric times through the Napoleonic era in 1975. While it covers both Eastern and Western civilizations, the narrative foregrounds European and Western developments, weaving together political, cultural, and intellectual histories with storytelling for a general readership (Durant, 1942). For detailed volume-specific distributions, see Figure 5 (temporal distribution of events) in the Appendix.

**The History of al-Tabari** (also known as *Tarikh al-Rusul wa al-Muluk*), compiled by Ibn Jarir al-Tabari and completed in 915CE, is an 11-volume annalistic chronicle beginning with creation and covering ancient empires, prophetic traditions, and Islamic history through to the early Abbasid caliphate. It offers an in-depth account of Middle Eastern history up to 915CE, with particular emphasis on Persian and early Islamic narratives (al Tabari & Rosenthal, 1988). The original text is in Arabic, and we conducted our analysis directly on the Arabic version to avoid potential issues introduced by translation nuances.

---

[1]We used three major historical works: *The Story of Civilization*, *The History of al-Tabari*, and *Man's Religions*, to enrich our dataset. No copyrighted text was reproduced; all historical content was paraphrased and fully attributed. This use aligns with standard academic fair-use (U.S.) and fair-dealing (U.K. and similar jurisdictions) practices, which permit paraphrasing factual material for non-commercial scholarly research provided attribution is given and no substantial portions of original expression are copied.

**Man's Religions** (by John B.Noss; revised edition c.1980s; originally early 1960s editions) is a single-volume comparative overview of global belief systems. It is organized in four thematic sections: primitive and extinct religions, religions of India, religions of East Asia, and religions of the Middle East, and provides factual, comparative descriptions of each tradition's history, beliefs, and practices (Noss, 1956).

Drawing on these sources and after the aggregation process, the resulting dataset includes **13,233 historical events**, categorized as follows: 11,176 from *The Story of Civilization*, 1,570 from *The History of al-Tabari*, and 487 from *Man's Religions*. The temporal distribution of these events is illustrated in Figure 2, which shows a higher density in the last 1,500 years. Each extracted event in our dataset is represented using the structured fields detailed in Table 2.

Table 2: Universal data schema for historical events

| Field | Description |
|---|---|
| **Event** | A short title or description of the event |
| **Source title** | Title of the event as it appears in the original text (if applicable) |
| **Time (original)** | Temporal description of the event as provided by the source |
| **Time (gregorian)** | Normalized year in the Gregorian calendar (negative for BCE, positive for CE) |
| **Time estimated** | Boolean flag: `true` if inferred, `false` if explicitly given in the source |
| **Place** | Name of the geographical location where the event occurred |
| **Place estimated** | Boolean flag: `true` if inferred, `false` if stated in the source |
| **Coordinates** | Standardized latitude and longitude of the location |
| **Main participants** | Key individuals or groups involved in the event |
| **Summary** | A concise summary of the event, optionally generated by a language model |
| **Source reference** | Name and volume of the source |
| **Page range** | Start and end pages of the event in the source material |

## 3.2 BENCHMARKING DATASET

To evaluate the performance of large language models (LLMs), we constructed a balanced benchmarking subset derived from our large-scale event dataset.

### 3.2.1 EVENT SELECTION

From the full corpus of 13,233 historical events, we selected a representative subset of 1,007 instances, ensuring coverage across diverse geographic regions, historical periods, and thematic domains. The dataset size was intentionally limited to a scale feasible for manual verification, thereby supporting the correctness and reliability of the benchmark. The distribution of the selected events is visualized in Figure 3, which demonstrates a similar distribution pattern between the full dataset and the benchmarking subset. Events from earlier historical periods are depicted in blue, transitioning to red for more recent events. Furthermore, areas with greater event density are represented with more intense colors, highlighting regions of significant historical concentration.

### 3.2.2 FACTUAL BENCHMARKING (LEVEL: REMEMBER)

Each of the 1,007 selected events was input into GPT-4o Mini to generate three multiple-choice questions, corresponding to the fields of time, place, and main participants. These questions were designed to assess the model's factual recall and knowledge retention. Only events that were answered correctly by all models across these three questions were retained for higher-level benchmarking.

### 3.2.3 HIGHER-ORDER BENCHMARKING VIA BLOOM'S TAXONOMY

To assess deeper historical reasoning beyond factual recall, we adopted Bloom's Taxonomy, a widely recognized framework for classifying educational learning objectives into six hierarchical cognitive levels (Anderson & Krathwohl, 2001). At the foundational level, *Remember* targets the retrieval of factual knowledge, such as dates, names, or specific events. The next level, *Understand*, involves grasping the meaning of historical content, such as summarizing a passage or interpreting a source. *Apply* requires learners to use historical knowledge in new contexts, for example, relating a past conflict to a contemporary situation. At a more advanced stage, *Analyze* focuses on breaking down historical narratives into components, identifying causes, effects, and relationships. The *Evaluate* level asks learners to make informed judgments, such as critiquing a historical decision or comparing the reliability of multiple sources. Finally, *Create* represents the highest cognitive level, involving the synthesis of new ideas or narratives based on historical understanding, such as constructing a counterfactual scenario or proposing an alternative interpretation of an event. This taxonomy informed the design of our evaluation framework, allowing us to probe different depths of reasoning, from simple recall to complex historical synthesis.

**Question Generation Process:** From the original set of 1,007 events, we first identified a subset of 394 events for which all tested models correctly answered the factual (i.e., "Remember" level) questions. For each of these events, we then generated five multiple-choice questions, each aligned with one of the higher-order levels of Bloom's Taxonomy: *Understand*, *Apply*, *Analyze*, *Evaluate*, and *Create*. The initial versions of these questions were produced using the GPT-4o Mini model. Subsequently, the questions were refined and their cognitive complexity enhanced using the DeepSeek model to ensure greater depth and challenge across the higher taxonomy levels.

This structured approach enables a comprehensive evaluation of LLMs across both lower and higher order cognitive skills in the domain of historical reasoning.

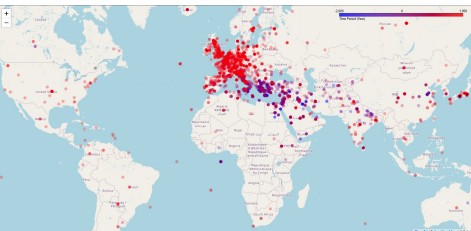 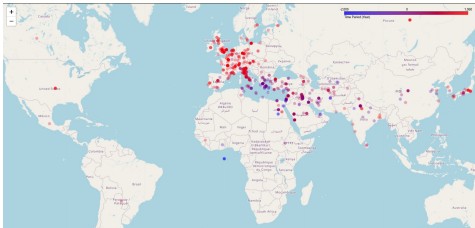

(a) Geographic distribution of the full event dataset.

(b) Geographic distribution of the benchmarking subset.

Figure 3: Comparison of the geographic distributions in the full dataset and the benchmarking subset. Time is visualized using a gradient from blue (older events) to red (more recent events). The density of events in each geographic area is represented by color intensity, highlighting historically rich regions.

## 4 METHODOLOGY

### 4.1 DATASET PREPARATION AND PREPROCESSING

We utilized digitized versions of three major historical texts: *The Story of Civilization* (Durant, 2016), *The History of al-Tabari* (al Tabari, 1967), and *Man's Religions*, the latter of which was digitized using Optical Character Recognition (OCR). Preprocessing involved cleaning the raw text

and segmenting each book into smaller, coherent chunks. Each chunk was given a descriptive title and annotated with its start and end page numbers, based on a structural analysis of the text.

## 4.2 EVENT EXTRACTION

We employed GPT-4 (32k context window) via prompt engineering to extract historical events from the preprocessed chunks. Two major challenges emerged in this process:

**(i) Missing temporal and spatial information:** In many cases, events lacked time or location data, both of which are essential for visualization on a temporal-spatial map. This issue stemmed either from limitations in the model's extraction capabilities or the absence of such details in the source text. To mitigate this, each prompt included both the target text segment and a set of recently extracted events to provide historical context. When time or place was not explicitly mentioned, the model was instructed to infer it based on its training data. A separate field was added to indicate whether this information was inferred (`True`) or directly stated (`False`).

**(ii) Standardization of extracted fields:** For consistency and usability, temporal data was converted into numeric formats (e.g., years, centuries), and spatial data into geographic coordinates (latitude and longitude). To support this, two additional fields were defined in the model prompt to extract standardized versions of time and location directly.

## 4.3 EVALUATION OF EXTRACTED EVENTS

To assess the quality of the extracted event dataset, a random sample of 50 events was selected for manual verification. Two independent evaluators reviewed each event's fields—including time, place, main participants, and others—labeling them as correct or incorrect based on careful examination of the original text and additional historical sources. Table 3 presents the results of this evaluation, including individual assessments and their average, demonstrating strong overall performance with an average accuracy of 94.1%. Notably, the standardization of place information exhibited slightly lower accuracy, reflecting challenges in precisely identifying geographical coordinates. These results indicate that the dataset is both robust and reliable for capturing critical historical event information.

Table 3: Evaluation of extracted events based on annotations by two dependent human annotators

|  | time | time estimated | time standard | place | place estimated | place standard | main participants | pages | total |
|---|---|---|---|---|---|---|---|---|---|
| annotator 1 | 90% | 96% | 98% | 94% | 100% | 88% | 96% | 100% | 95.25% |
| annotator 2 | 88% | 96% | 88% | 94% | 98% | 88% | 98% | 94% | 93% |
| average | 89% | 96% | 93% | 94% | 99% | 88% | 97% | 97% | 94.125% |

## 4.4 LLM EVALUATION

We evaluated the performance of three large language models: GPT-4o Mini (OpenAI et al., 2024), DeepSeek-V3 (DeepSeek-AI et al., 2025), and Gemma-3-27B-IT (Team et al., 2025). Evaluation proceeded in two stages:

**Factual Benchmarking (Remember level):** Each model was assessed using three multiple-choice questions per event, targeting the fields of time, place, and main participants.

**Higher-Order Reasoning Benchmarking:** Events for which all three models answered correctly at the factual level were selected to generate more advanced questions. These were mapped to the upper levels of Bloom's Taxonomy (*Understand*, *Apply*, *Analyze*, *Evaluate*, and *Create*) to evaluate the models' deeper historical reasoning capabilities.

## 5 RESULTS

For each multiple-choice question, the model's response was evaluated against the ground truth to determine its correctness. Overall accuracy was then calculated based on the proportion of correct responses. Table 4 presents the performance of the three models at the *Remember* level, while

Table 5 reports their results across the remaining five levels of Bloom's Taxonomy. They offer a detailed view of how different large language models perform across various dimensions of historical understanding. Below are several key insights drawn from the evaluation data:

**Overall Performance Levels.** **(1)** All models generally perform better on higher-order cognitive tasks (like *Evaluate* and *Create*) compared to the *Remember* and *Understand* levels. **(2)** *DeepSeek-V3* consistently outperforms *GPT4o-mini* and *Gemma-3* across nearly all categories and Bloom's levels, indicating stronger historical reasoning and comprehension capabilities.

**Remember Level (Table 4).** **(1)** Models excel in recognizing *Place* and *Main Participants*, with accuracy around 90% or above, while performance on *Time* is considerably lower (66.5%–75.9%). This suggests temporal understanding remains more challenging than spatial or entity recognition at the factual recall level. **(2)** *DeepSeek-V3* leads on all three *Remember* subcategories, pushing its total accuracy to 88.65%, about 5 percentage points higher than the other two models.

**Higher-Order Cognitive Levels (Table 5).** **(1)** Accuracy improves progressively from *Understand* (approximately 74–84%) to *Evaluate* and *Create* levels (approximately 79–92%), demonstrating that models can perform well on complex reasoning tasks when provided with structured historical data. **(2)** *DeepSeek-V3* again ranks highest across all five levels, exceeding 90% accuracy at *Evaluate* and *Create*, suggesting a better grasp of complex historical concepts and analysis. **(3)** *Gemma-3* trails behind *GPT4o-mini*, especially at the *Apply*, *Analyze*, *Evaluate*, and *Create* levels, indicating weaker performance in applying and synthesizing historical information.

Table 4: Model performance at the remember level, showing the number of correct answers alongside the corresponding accuracy percentages

| model | time | place | main participants | total |
|---|---|---|---|---|
| gpt4o-mini | 670 (66.534%) | 911 (90.466%) | 943 (93.644%) | 83.55% |
| deepseek-v3 | 764 (75.868%) | 955 (94.836%) | 959 (95.233%) | 88.65% |
| gemma-3-27b-it | 704 (69.911%) | 911 (90.466%) | 916 (90.963%) | 83.78% |

Table 5: Model performance on higher-order levels (bloom's taxonomy), showing the number of correct answers alongside the corresponding accuracy percentages

| model | understand | apply | analyze | evaluate | create |
|---|---|---|---|---|---|
| gpt4o-mini | 298 (75.63 %) | 327 (82.99 %) | 348 (88.32 %) | 357 (90.60 %) | 349 (88.57 %) |
| deepseek-v3 | 332 (84.26 %) | 335 (85.02 %) | 351 (89.08 %) | 364 (92.38 %) | 362 (91.87 %) |
| gemma-3-27b-it | 291 (73.85 %) | 301 (76.39 %) | 312 (79.18 %) | 327 (82.99 %) | 311 (78.93 %) |

# 6 VISUALIZATION

To facilitate the exploration of the extracted historical events, we developed a web-based visualization platform featuring an interactive 3D globe. Users can select specific time intervals, by year or century, and view the corresponding events geographically displayed on the globe. An adjustable timeline is provided to further refine the temporal range and dynamically update the displayed events.

The intensity of the color bars on the map increases with the number of events associated with a given location; higher event density results in more saturated color markers. By hovering over a location, users can access a tooltip displaying detailed information about the associated events.

Additionally, a side panel presents a scrollable list of all currently filtered events, allowing for easier navigation and inspection.

This visualization platform is implemented using HTML and JavaScript, with the support of the Globe.GL library [2], a UI component built on Three.js/WebGL for interactive geographic data visualization. A screenshot of the interface is shown in Figure 4.

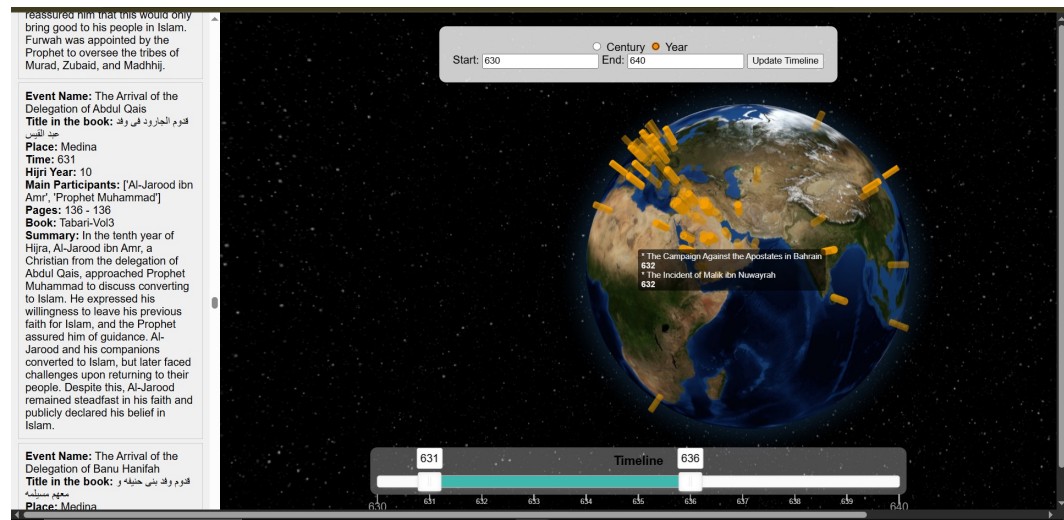

Figure 4: An example visualization of historical events on the interactive globe interface.

## 7 CONCLUSION

This paper introduced *HistoBench*, a comprehensive benchmark and dataset aimed at evaluating large language models' (LLMs) capabilities in understanding temporally grounded and context-rich historical narratives. By extracting and structuring over 13,000 events from diverse historical texts, we not only broadened the geographic and cultural scope of available historical datasets but also enabled meaningful analysis through an interactive globe-based visualization interface. Furthermore, we constructed a cognitively balanced benchmark of 1,007 multiple-choice questions, guided by Bloom's Taxonomy, to assess both factual recall and higher-order reasoning in history-focused tasks.

Our evaluation of three leading LLMs revealed notable performance differences across cognitive levels and question types, with DeepSeek-V3 demonstrating superior accuracy and reasoning consistency. These findings highlight both the potential and current limitations of LLMs in processing complex historical content. Further work may explore expanding the dataset to cover a broader range of cultures and historical traditions, as well as extracting additional layers of information, such as historical figures, their relationships, and interconnections, to enable more advanced forms of contextual and relational reasoning in historical language understanding.

---

[2]https://globe.gl/

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

# A APPENDIX

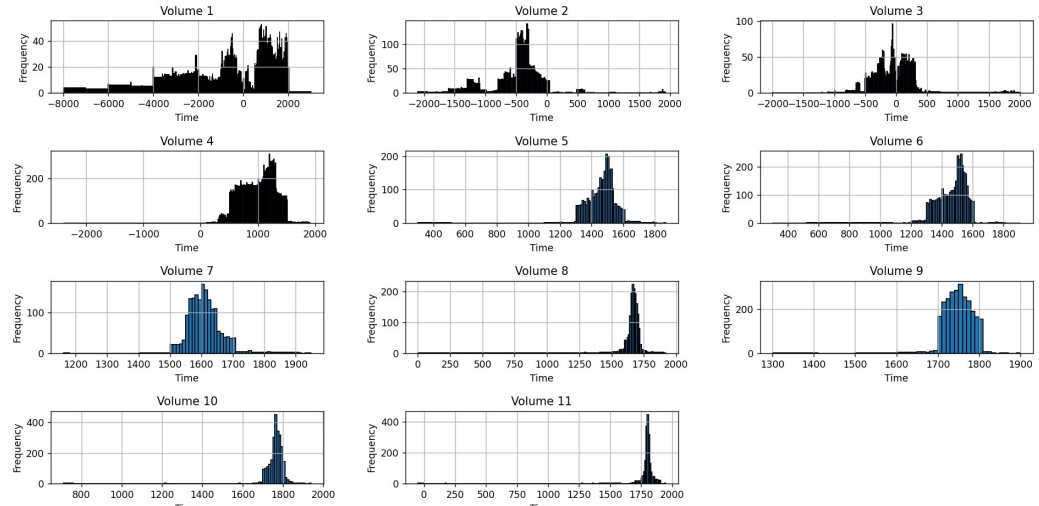

Figure 5: Bar chart showing the temporal distribution of extracted events by volume.

