# OpenReview forum: "HistoBench: World History Event Extraction and Cognitive-Level Benchmarking of Generative AI"
_ICLR.cc/2026/Conference — ICLR 2026 Conference Withdrawn Submission_

### Official Review · Reviewer_P8cL · 2025-10-31

**Soundness:** 2
**Presentation:** 3
**Contribution:** 1
**Rating:** 2
**Confidence:** 4

**Summary:**

This paper presents HistoBench, a benchmark for evaluating factual recall and complex reasoning of LLMs regarding historical events.
Overall, 13k historical events are extracted from 3 books. For each event, the event name, location and date are retrieved or inferred. A subset of 50 randomly selected events are manually checked.

A benchmark is created using 1007 of those events. It has the form of multi-choice questions, evaluates simple recall, and contains three questions per event (regarding the location, the time and the main participants). Three models are evaluated with this benchmark: gpt4o-mini, deepseek-v3 and gemma-3. Results show that the models mainly struggle in time related recall questions.

For the events known by all three models (comprising 394 events), an additional benchmark is created. This benchmark follows the Bloom's taxonomy and is separated into 5 multi-choice question/answer datasets, corresponding to 'understand', 'apply', 'analyze', 'evaluate' and 'create'. No additional information is provided about the construction of those additional benchmarking datasets. Results show that deepseek-v3 consistently outperforms the other two models.

Finally, a visualization platform is provided.

**Strengths:**

- full process for extracting the events from cleaning the raw data to visualization.
- extracting and automating the events in a structured way is interesting

**Weaknesses:**

- For the event extraction part, there are a lot of missing explanations: how the chunks have been selected? How the events have been extracted? What are the prompts used (there is no prompt provided)?
- For the benchmarking part, there is no explanation at all regarding the creation of the benchmarks. In particular: there is no provided example of a multi-choice question. Additionally, I don't see how complex-reasoning questions related to e.g. "constructing a counterfactual scenario" can be evaluated with a multi-choice question produced by gpt-4o-mini and deepseek-v3.
- The evaluation is shallow (three models evaluated). Some evaluated models have also created the questions (in particular, gpt4o-mini and deepseek have been used for creating and refining the most complex questions)
- There is no explanation about the difference of performance for time w.r.t the other components evaluated (place and participants).

**Questions:**

Q1. Is the full process automatized? Is it possible to scale it? What is the expected time to process and extract the events of an additional book?

Q2. Can you provide examples of each benchmark?

Q3. In Fig. 3, some selected events are located in the null island. Have those been filtered?

Q4. The results show almost saturated performance (90%+) on the most complex case ('create') with deepseek-v3. What is the purpose for the research community: evaluating small LLMs in particular for a specific objective?

---

### Official Review · Reviewer_sTCr · 2025-10-31

**Soundness:** 1
**Presentation:** 1
**Contribution:** 1
**Rating:** 0
**Confidence:** 5

**Summary:**

this paper proposes a new benchmark for historical questions: its main contribution is in the procedure to create the benchmark from 3 historical sources.

the paper has several problems, first and foremost a likely flawed extraction methodology that makes the whole dataset, and associated evaluation, a very weak contribution.

**Strengths:**

- unclear

**Weaknesses:**

- NLP incoherence
- flawed methodology
- biased final corpus
- irrelevant resutls
- unclear cost

**Questions:**

## NLP incoherence

author staate they do not translate

		``The original text is in Arabic, and we conducted our analysis directly on the Arabic version to avoid potential issues introduced by translation nuances.''

but then they have to anyway paraphrase due to copyright issuses

	``No copyrighted text was reproduced; all historical content was paraphrased and fully attributed.''

how is paraphrasing different than translation for losses of nuance?
accessorily, how robust is GPT-4 knowledge of arabic



## flawed methodology ?


I have serious doubts concernin the soundness of the processing. Taking one of the sources The Story of Civilization  (1975) 	Volume I appears to tackle a significant fraction of events from recent years (your Appendix A)

By cursory analysis of the dates (East asia, Egypt) the book stops before Christ in Japan before WWII (1935: Notice given to terminate Washington Agreement) since that is the year the fisrt volume was written -- Considering the preface is signed "WILL DURANT. Great Neck, N. Y., March, 1935"


Your plot in the top right cornet is therefore highly suspicious, as in no other book the distruibution of the events also cover up to year 2000 (and even after that!) while the first volume would have to stop at 1935 and the full series in 1975.

this fact alone, oif processing relatively straigthforward data (eg timestamp) can lead to uncertain results, then this casts doubts on the whole processing

prompts and code should be made available and duly explained in appendix material.


## events under-representation

while your goal is representativity, I believe the events in the 13k events corpus are grossly undercountend -- in a way that is not controlled, and therefore introducing an unknown and unmeasured bias

considering the number of events, 13,000 events from 3 books seems an undercounting. Even taking just volume 1 of the 11-volume series of book #	The Story of Civilization  (1975), the index has 47 pages (pp1002-1049) with roughly 100x events each, so about  5000 names mentioned in just book.
this is not my understanding looking at the index of the book https://dn720004.ca.archive.org/0/items/TheStoryOfCivilizationcomplete/Durant_Will_-_The_story_of_civilization_1.pdf#page=1150.17


Such entities can be places, battles, people and some places  (eg Rome) and entities (eg Theodore Roosvelt)  can  be mentioned several times:

Rome, 3, 19", 24",61,76, I16, I17, 136, 140,
152, 172, 185, 200, 216, 218, 226, 227, 247,
265, 27 2, 275, 284, 299, 315. 340, 354, 362 ,
363, 381 -3 82 , 45 1, 479, 529, 554, 640, 647.
695. 701 , 744, 777, 778, 847, 899, 925
Rome (city), 155, 294, 457

Roosevelt, Theodore, President of the
United States (1858-1919), 918, 929-930

How to understand which events are missed, and so which bias is introduced, is not an easy challenge.


## biased output corpus

Letting the above aproblem aside

" From the full corpus of 13,233 historical events, we selected a representative subset of 1,007 instances, ensuring coverage across diverse geographic regions"

howerver, how representative is the subset of 1000 out of 13k ?
	I agree North america is over-represented in LLMs, but From  Fig 3  it seems to be  is heavily
	undersampled in your final selection.  	No metric of geographic coverage, or mention of stratified seleciton appears to support the selection representativeness .

## lack of statistical rigous

The paper present a candid lack of statistical rigour:   "a random sample of 50 events" 	there is no statistical relevance of the study from 2 annotators that can be projected to the corpus of 1000  events selected out of the 13,233 events.

given the above apparently quite large gaps from expectations (# of events, timeline/dates, geographic coverage) this makes it difficult to assess the value of the work, with respect to say HistBench (without "o")

## irrelevant results

all models look quite close in performance and given the above problems, what is that we learn from HistoBench that cannot be learned from HistBench ?

Is there any difference rooted in the Bloom taxonomy, or stratified selection that lead to different conclusions ?




##  unclear cost

 we employed GPT-4  -- what was the overall processing cost?

---

### Official Review · Reviewer_kpkQ · 2025-10-31

**Soundness:** 2
**Presentation:** 3
**Contribution:** 2
**Rating:** 2
**Confidence:** 4

**Summary:**

This paper introduces a new benchmark that covers historical event understadning. Authors describe the data creation process and benchmark frontier and open-weight LLMs on it.

**Strengths:**

- The main contribution is the proposed benchmark itself

**Weaknesses:**

Overall I find that this paper lacks scientific rigor and in-depth analysis. More specifically:

**[W1]** The authors note that certain events are underrepresented, but do not clarify how their benchmark overlaps with existing benchmarks that test historical and general factual knowledge, such as MMLU. It is also unclear how their benchmark correlates with the Wikipedia corpora commonly used for pre-training. Additionally, it is not specified whether the books forming the basis of this benchmark are already included in existing pre-training datasets. This raises questions about the novelty and uniqueness of the events present in the proposed benchmark.

**[W2]** There are numerous design choices in the data creation process that are not described in the paper. For further details, please refer to my specific questions in the questions section.

**[W3]** The paper evaluates only three models, which limits the generalizability of the findings. Furthermore, many important details regarding the evaluation process are missing. Please see my questions in the next section for specifics. Without this information, it is impossible to properly assess the validity of the results and the overall contribution of the work.

**[W4]** The analysis provided is limited. For example, questions categorized under "Understand" exhibit the lowest performance, even compared to more complex reasoning categories. This contradicts the authors' stated motivation that resources are needed not just for memorizing isolated facts, but for understanding complex relationships between events, people, and time periods.

**Questions:**

Related to [W2]
- Why were these specific books chosen as the starting point for data creation?
- How did the authors extract events from these books?
- How do the authors define an "event"? Given the extensive literature on event extraction and temporal standardization over the past decade, what is the motivation for not leveraging existing methodologies to extract and categorize these timelines? How do the authors address different levels of event granularity and event overlapping?
- What prompts were used at each preprocessing stage?
- What rubric and verification criteria were used for human annotation?
- Why did the authors not validate all extracted events? With only 1,007 events, manual inspection seems feasible, especially since the benchmark, which is the main contribution of this work, is based on these events.
- Can you provide question example of the proposed benchmark for each category?

Realted to [W3]:
- What evaluation metric is used for the non-MCQ part of the benchmark?
- What is the evaluation setting? few-shot or zero-shot?
- What generation instructions were provided (e.g., direct answer or chain-of-thought)?

---

### Note · Authors · 2025-12-02

I have read and agree with the venue's withdrawal policy on behalf of myself and my co-authors.